# Trajectory patterns of SARS-CoV-2 neutralising antibody response in convalescent COVID-19 patients

Ngai Sze Wong [1,2,6], Shui Shan Lee [1,6], Denise P. C. Chan[1], Timothy C. M. Li[3], Tracy H. Y. Ho[3], Fion W. L. Luk[3], Kai Ming Chow [3], Eugene Y. K. Tso[4], Eng-Kiong Yeoh[2,5], Samuel Y. S. Wong[2], David S. C. Hui [3] & Grace C. Y. Lui [3✉]

## Abstract

**Background** The adaptive immune responses of COVID-19 patients contributes to virus clearance, restoration of health and protection from re-infection. The patterns of and the associated characteristics with longitudinal neutralising antibody (NAb) response following SARS-CoV-2 infection are important in their potential association with the population risks of re-infection.

**Methods** This is a longitudinal study with blood samples and clinical data collected in adults aged 18 or above following diagnosis of SARS-CoV-2 infection. NAb levels were measured by the SARS-CoV-2 surrogate virus neutralisation test (sVNT). Anonymous clinical and laboratory data were matched with surveillance data for each subject for enabling analyses and applying latent class mixed models for trajectory delineation. Logistic regression models were performed to compare the characteristics between the identified classes.

**Results** In 2020–2021, 368 convalescent patients in Hong Kong are tested for NAb. Their seroconversion occur within 3 months in 97% symptomatic patients, the level of which are maintained at 97% after 9 months. The NAb trajectories of 200 symptomatic patients are classified by the initial response and subsequent trend into high-persistent and waning classes in latent class mixed models. High-persistent (15.5%) class patients are older and most have chronic illnesses. Waning class patients (84.5%) are largely young adults who are mildly symptomatic including 2 who seanorevert after 10 months.

**Conclusions** Characteristic sub-class variabilities in clinical pattern are noted especially among patients with waning NAb. The heterogeneity of the NAb trajectory patterns and their clinical association can be important for informing vaccination strategy to prevent re-infection.

## Plain language summary

Neutralising antibodies are produced by the immune system and help to defend against viruses like SARS-CoV-2, which causes COVID-19. Declining levels of these antibodies over time might be linked to risk of re-infection with the virus. Here, we look at changes in neutralising antibody levels over time in people who have had COVID-19. We define two classes of people: those with persistently high levels of antibodies over time, who are more likely to be older and have chronic illnesses, and those with declining antibody levels, who are younger and had mild COVID-19 symptoms. Understanding differences in how these antibodies are maintained over time in different groups of people might help to guide vaccination strategies to prevent re-infection.

[1] Stanley Ho Centre for Emerging Infectious Diseases, The Chinese University of Hong Kong, Hong Kong, Shatin, China. [2] JC School of Public Health and Primary Care, The Chinese University of Hong Kong, Hong Kong, Shatin, China. [3] Department of Medicine and Therapeutics, The Chinese University of Hong Kong, Hong Kong, Shatin, China. [4] Department of Medicine & Geriatrics, United Christian Hospital, Hong Kong, China. [5] Centre for Health Systems and Policy Research, The Chinese University of Hong Kong, Hong Kong, Shatin, China. [6] These authors contributed equally: Ngai Sze Wong, Shui Shan Lee. ✉email: gracelui@cuhk.edu.hk

SARS-CoV-2, the causative virus of COVID-19, is an extremely contagious pathogen that has continued to cause major outbreaks globally ever since its emergence in late 2019 (https://www.who.int/emergencies/diseases/novel-coronavirus-2019). Before the widespread use of vaccine and in the absence of definitive curative therapy in most places, recovery of SARS-CoV-2 infected patients depends almost exclusively on the functioning of host immunity[1]. Studies have shown that both humoral and cell-mediated immunity contributed to the clearance of the virus and restoration of health[2–4]. With the increasing availability of validated serological tests, antibody responses have become widely investigated for clinical evaluation. The profiling of antibody responses covers different antigen and immunoglobulin classes. Whereas different components of humoral immunity exhibited distinct kinetics[5], antibody against receptor binding domain (RBD) or spike protein normally rises to a peak and becomes maintained for over 5 months, corresponding with virus neutralisation capacity[6]. Similar pattern was observed in another study on seropositive healthcare workers with antibody monitoring for 8 months after symptom onset[7].

An evaluation of the dynamics of neutralising antibody responses after acute SARS-CoV-2 infection is crucial for understanding not just the pace of health recovery but also the risk of re-infection. A study reported that neutralising antibody progressively falls after 5–8 weeks after symptomatic infection though it remains detectable by 8 months[8]. Asymptomatic and mildly symptomatic infections are associated with lower neutralising antibody levels[9,10], while older patients may have higher level[10]. Overall, heterogeneity of neutralising antibody responses within 3 months after diagnoses has been shown in some studies[10,11]. A more recent study with sampling up to 6 months has identified 5 different patterns of neutralising antibody dynamics[12]. With vaccination playing a central role in protecting individuals from SARS-CoV-2 infection, delineation of the dynamics of neutralising antibody response over longer interval would be important in assessing the vaccination needs of convalescent patients. It is against this background that we undertook to examine the trajectory of neutralising antibody (NAb) responses and factors associated with its pattern in SARS-CoV-2 patients who had recovered from COVID-19 of different severity. In this study, we classified convalescent symptomatic patients by their differences in the intensity of NAb responses initially and overtime, showing that high-persistent responses were associated with older age and the presence of chronic illnesses.

## Methods

**Study design and data sources.** This is a longitudinal study with the collection of blood samples and clinical data in COVID-19 patients between February 2020 and February 2021 in Hong Kong, where stringent isolation policy is in place with all confirmed cases are reported and hospitalised irrespective of symptoms, travel, and contact history. COVID-19 vaccination programme started from 22 February 2021 while the reporting date of the last recruited patient was 20 February 2021. Blood sampling was performed at multiple time points during hospitalisation and follow-ups after discharge. Written informed consent was obtained from each participant. Separately, anonymous surveillance data of COVID-19 reported cases in Hong Kong were collected from the Centre for Health Protection, Hong Kong Special Administrative Region Government. The data included socio-demographics, history of chronic illnesses at diagnosis, travel history, and epidemiological linkages[13]. Anonymous clinical and laboratory records of all reported cases with case identifiers matched with surveillance data were retrieved from the Hospital Authority, Hong Kong. Ethical approval of the Joint Chinese University of Hong Kong—New Territories East Cluster

Clinical Research Ethics Committee (CREC Reference Number: 2020.218) was obtained. This study is registered on Clinical-Trials.gov (NCT05028881).

**Participants and setting.** Patients were recruited from the Prince of Wales Hospital, a tertiary hospital in the public service with catchment for some 1.8 million population in the New Territories East Region of Hong Kong. The inclusion criteria included adults of age 18 or above admitted to the hospital with a confirmed diagnosis of SARS-CoV-2 infection, with the detection of SARS-CoV-2 nucleic acid from respiratory specimens. Diagnosis of SARS-CoV-2 infection was made by reverse transcription polymerase chain reaction (RT-PCR) assays on respiratory specimens at the hospital laboratory followed by confirmation by the Public Health Laboratory of Centre for Health Protection.

**COVID-19 serology.** The humoral response against SARS-CoV-2 including IgG and neutralising antibodies were examined by using commercially available ELISAs. The analysis of SARS-CoV-2 antibodies were performed using the SARS-CoV-2 NP IgG ELISA (nucleocapsid protein-based antigen; ImmunoDiagnostics) for IgG NP, and anti-SARS-CoV-2 IgG ELISA (S1 subunit of spike protein-based antigen; EUROIMMUN) for IgG spike. NAb were measured by the SARS-CoV-2 surrogate virus neutralisation test (sVNT), based on antibody-mediated blockage of ACE2-spike RBD interaction (RBD-targeting NAbs; GenScript)[14]. A positive NAb result with sVNT was defined as a level above 20% inhibition based on manufacturer's recommendation. The tested specimens were heparinised blood which were separated and stored in aliquots of 0.5 mL at −80 °C until use. They were retrieved and heat-inactivated at 56 °C for 30 mins before testing. All assays were performed in accordance with the manufacturers' instructions.

**Data processing and analysis.** Patients with both IgG NP and NAb measured between day 15 and 90 from onset date for symptomatic patients or reporting date for asymptomatic patients were included in the study. The main outcome was NAb level, and the secondary outcomes were IgG spike and IgG NP detection. We defined severe/critical condition when there was diagnosis of pneumonia and requirement of supplemental oxygen, or organ support, or admission to intensive care unit (ICU). The sociodemographic variables including gender, age, onset date, reporting date, hospital admission and discharge dates, travel history (e.g. origin and mode of travel), and epidemiological linkage collected in this study were used to link the surveillance and hospital data. Records of patients that could not be successfully linked with other data sources were included in the analyses, with the corresponding variables marked as missing values. Descriptive statistics were used to display the characteristics of the data in the study.

To delineate the trajectories of NAb (ranging between 0% and 100%) since the onset date, symptomatic participants with at least 2 sVNT measurements in separate months and who had been followed-up for at least 3 months from the onset date were analysed. The time variable in the analysis was the interval (months) between the onset date and the blood sample collection date. During hospitalisation, the first record, and records with the maximum and the minimum values of repeat laboratory measurements were selected as variables. Laboratory markers included albumin, CKD-EPI eGFR[15], Ct-value of SARS-CoV-2 RT-PCR from upper respiratory tract specimens, globulin, haemoglobulin, lymphocyte, neutrophil, and white blood cell counts.

Latent class mixed model (LCMM) was applied for NAb trajectory delineation. LCMM is a combination of latent class model and mixed model for repeated measurements, which have

been applied in classifying the anti-spike antibody response to SARS-CoV-2 vaccines[16], and natural SARS-CoV-2 infection[17]. In this study, we performed LCMM in lcmm R package[18]. Both linear and non-linear functions (beta, 3-quant-splines, 5-quant-splines, and custom splines) were performed and compared. Model with lower Bayesian information criterion (BIC) value was selected. Based on posterior probability of ≥70%, patients were assigned to the class with the highest probability. To compare the characteristics of patients between identified classes, bivariable logistic regression models were performed in SPSS 25[19]. In light of the potential association of age and chronic illnesses, we included the variable of age 60 or above as a confounder in multivariable logistic regression models. Multivariable multi-nominal logistic regression models were performed to compare more than two classes. Complete-case analyses were performed.

**Reporting summary**. Further information on research design is available in the Nature Research Reporting Summary linked to this article.

## Results

**General characteristics**. Between February 2020 and February 2021, a total of 448 patients were recruited in the study. Of these, 368 with IgG NP, IgG spike and NAb measurements in day 15–90 were selected for analysis. Overall, 51% were male, 28% aged 60 or above (median 48 years old (yo), IQR 31-61 yo) (Supplementary Data 1). Seven percent ($n = 27$) were asymptomatic, and none meeting the inclusion criteria have died. A total of 126 cases (34%) gave a history of chronic illnesses, more than half (67/126, 53%) of which with cardiovascular diseases, and about half (60/126, 48%) with metabolic disorders. Some 14% (50/368) of the patients had hypertension, diabetes mellitus and/or hyperlipidaemia (HT/DM/HL) only, 11% reported other chronic illnesses while 14% had HT/DM/HL in conjunction with other chronic illnesses. A higher proportion of patients aged 60 or above had multiple chronic illnesses, experienced severe/critical conditions, lower Ct values and biochemical markers reflective of poorer clinical status (Supplementary Data 2).

We defined seroconversion as the detection of antibody against SARS-CoV-2. For symptomatic cases, seroconversion was documented in 361/368 (98%), 158/162 (98%) and 355/365 (97%) for IgG NP, IgG spike and NAb respectively. With the exception of 2 cases (1 for IgG NP and 1 for NAb), seroconversion occurred within 3 months after symptom onset (Supplementary Data 3). The proportion of asymptomatic patients who tested positive for IgG NP (81%) and NAb (81%) was lower than symptomatic patients (IgG NP 99%; NAb 98%) in the first 3 months after initial diagnosis. Seroconversion status could not be substantiated in 4 patients tested negative for any of the 3 antibody markers within 3 months and were lost to followup. The proportion with positive results with NAb and IgG NP was high at 97% and 94% respectively after 9 months, while that for IgG spike dropped after month 3 to 83% through 9 months among symptomatic patients. The NAb results from sVNT were expressed as percentage inhibition for comparison. Overall, a peak value could be obtained within 90 days after symptom onset (Fig. 1), with the maximum NAb value of 98%. The median NAb per month were high at >70% for readings between 3 and 6 months (except month 4 with very limited sample size), but were low at <60% for timepoints at and after month 9 (day 270).

**Neutralising antibody trajectories**. A total of 200 symptomatic patients each with 2 to 5 NAb measurements spanning between day 15 to month 14 from the onset date were classified. The median follow up duration was 6 months (IQR 5–9 months). In the first 3 months, around two-thirds of the NAb values were ≥80%, and 16% were 45–60% (Fig. 2). NAb readings in the following half year (months 3–9) were in relatively even distribution ranging between 20.3% and 98.1%. From month 9 onwards, around half (49%) of the NAb readings ranged 30–60%.

With time as the variable for the fixed effects and random effects in the mixed model and as a fixed effect in the class-membership model, using 5-equi-splines (5 equidistant nodes) non-linear function in LCMM, 2 main NAb trajectory classes were identified (BIC = 3838.55, Supplementary Data 4): "high-persistent" (Class 1, $n = 31$), "waning" (Class 2, $n = 169$) (Fig. 3). In the first 3 months, the median baseline NAb in high-persistent class was 95% (IQR 93–97%), significantly different from that of the waning class at 85% (IQR 72–92%) (Mann–Whitney $U = 891$, $p < 0.001$) (Supplementary Fig. 1). The corresponding distribution of IgG spike by class is shown in Supplementary Fig. 2. Towards the end of one year (>9 months), the high-persistent class continued to have high median level at 94% (IQR 93–97%), but that of the waning class dropped to 47% (IQR 37–64%) (Mann–Whitney $U = 0$, $p < 0.001$). Overall, only 2 patients (male, age 64 and 23) seroreverted after 10 and 12 months from onset, and both belonged to the waning class.

Compared to the waning class, a significantly higher proportion of patients in the high-persistent class were aged 60 or above (odds ratio (OR) = 4.36, 95% C.I. = 1.96–9.7) (Supplementary Data 5). Adjusted for aged 60 or above in multivariable logistic regression model, high-persistent class patients were more likely to have chronic illnesses (adjusted OR (aOR) = 3.06, 95% C.I. = 1.30–7.25), both HT/DM/HL and other chronic illnesses (aOR = 5.23, 95% C.I. = 1.80–15.19; no chronic illnesses reported as reference), a history of cardiovascular diseases (aOR = 4.47, 95% C.I. = 1.85–10.82), metabolic diseases (aOR = 2.99, 95% C.I. = 1.26–7.12) or liver diseases (aOR = 6.85, 95% C.I. = 1.32–35.49). Presence of severe/critical conditions (OR = 3.18, 95% C.I. = 1.32–7.64; aOR = 2.08, 95% C.I. = 0.82–5.31) and having received COVID-19 treatment (OR = 2.45, 95% C.I. = 1.0001–6.02; aOR = 1.95, 95% C.I. = 0.77–4.95) were significant factors for being high-persistent class in bivariable logistic regression but were insignificant after adjusting for age group. Clinically, there were significant between-class differences in the baseline characteristics and in-hospital biochemical and haematological markers before but not after adjustment of age group as a confounder (Supplementary Data 5). The SARS-CoV-2 viral load as inferred from Ct values in our cohort was similar between classes, but patients aged 60 or above as a group tended to have lower Ct at diagnosis.

As the waning class accounted for 84.5% of patients, characteristics of patients in 3 classes and 4 classes LCMM were further examined in sensitivity analysis, despite higher BIC (3843.73 for 3 classes and 3847.03 for 4 classes) and smaller sample size per class (Supplementary Fig. 3). The NAb trajectory pattern were classified by the initial NAb response (high, moderate) and subsequent trend (persistent or waning) in a 3-class model: modified Class 1 "high-persistent" ($n = 31$), Class 2A "high-waning" ($n = 94$), and Class 2B "moderate-waning" ($n = 75$). Age has remained significantly different between classes, with modified Class 1 being older (median 63 yo), followed by Class 2A (median 55 yo), and younger in Class 2B (median 37 yo). Comparing the characteristics between modified Class 1 with Class 2A and 2B separately, the associated factors were similar to those in the 2-class model, except that the history of metabolic or liver diseases were no longer significant in multivariable multi-nominal logistic regression models (Supplementary Data 6). None of the cases in Class 2B had presented with severe/critical

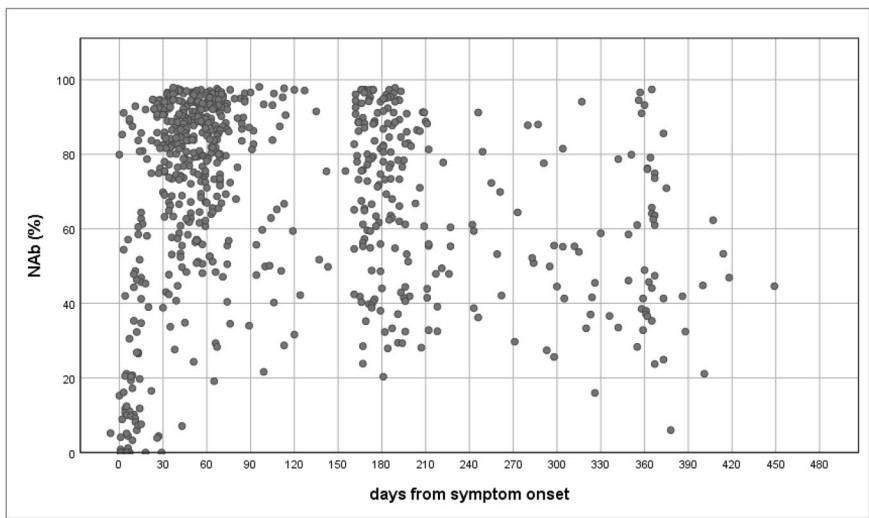

**Fig. 1 Scattered plot for neutralising antibody (NAb) levels (%) over days from symptom onset among 341 symptomatic patients.** NAb—neutralising antibody.

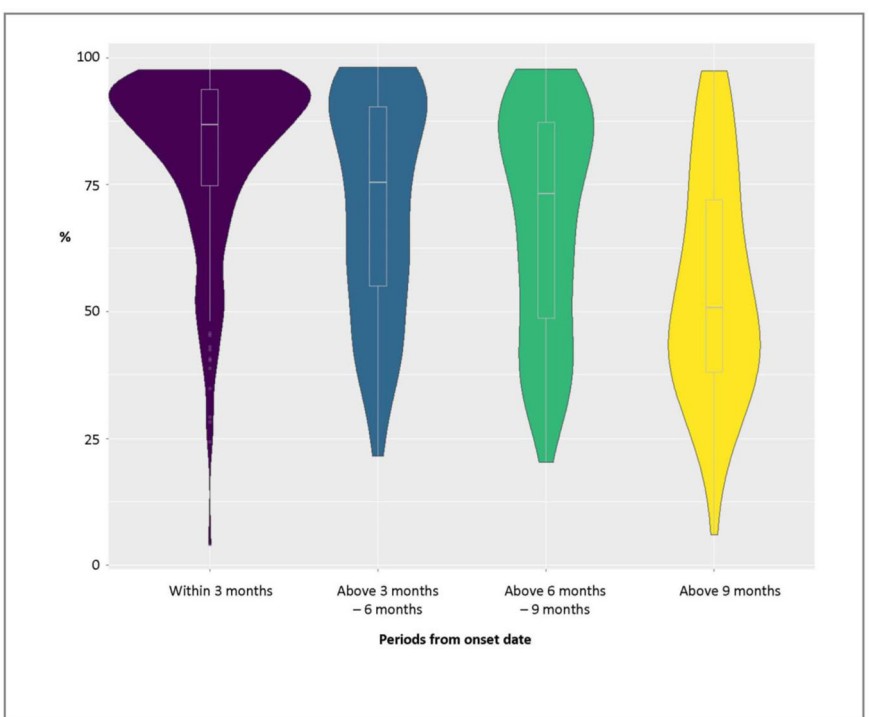

**Fig. 2 Distribution of neutralising antibody (NAb) levels in % by surrogate virus neutralisation test (sVNT) in different periods from the onset date, among 490 measurements in 200 patients.** Violin plot with box plot (light grey marker for median and box for interquartile range) showing the distribution of neutralising antibody (NAb) levels in % by periods from the onset date: within 3 months (208 measurements), above 3 months – 6 months (110 measurements), above 6 months – 9 months (99 measurements), and above 9 months (73 measurements); note: 1 patient may have more than one measurement in the same period.

conditions. In addition, a number of biochemical and haematological markers were significantly different between modified Class 1 and Class 2. Characteristically, modified Class 1 gave a lower albumin and haemoglobulin on admission, compared to Class 2B. During hospitalisation, Class 1 patients had higher maximum globulin, lower minimum albumin, eGFR, haemoglobulin, and lymphocyte than Class 2B, after adjustment by age group. Similar findings were observed for patients classified into 4 classes (Supplementary Data 7–8).

## Discussions

From our analyses on adults recovering from SARS-CoV-2 infection, high-persistent and waning patient classes could be delineated by the NAb trajectory pattern using sVNT for quantifying the neutralising capacity of antibody developed in response to the infection[14]. Notably, about 15% began with high NAb level and persisted in the course of 1 year (high-persistent class), while the rest showed waning NAb. The main factors for delineating the classes were age and pre-morbid medical

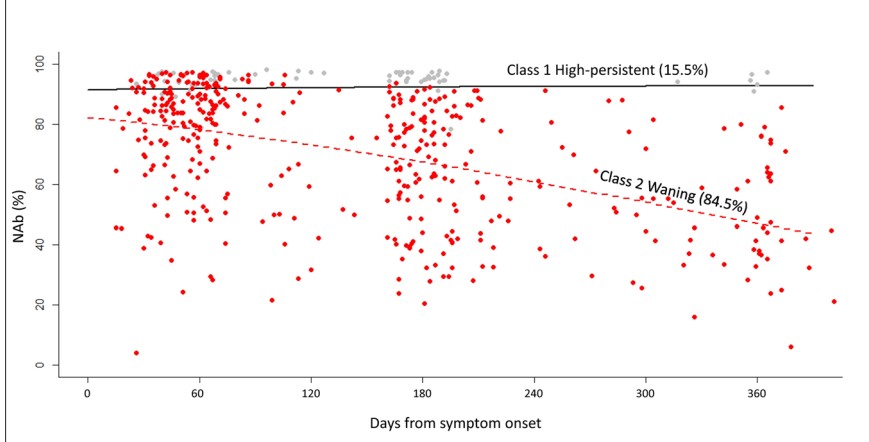

**Fig. 3 Class-specific mean predicted trajectory of neutralising antibody (NAb) level by surrogate virus neutralisation test (sVNT), _n_ = 200.** Trajectory of neutralising antibody (NAb) level for Class 1 (31 patients) and Class 2 (169 patients).

condition. The waning class represented a majority of patients who were otherwise healthy adults before the outbreaks. Most of them did not achieve a very high NAb level, and a gradual decline was noted and maintained at a median level below 50% after 1 year. A low level of NAb had been shown in previous studies on asymptomatic patient[9,10].

Accounting for 15.5% of the study population, the high-persistent class in the 2-class model comprised mainly elderly patients with chronic illnesses who, following SARS-CoV-2 infection, could generate high levels of NAb. Our results echoed that of a shorter 6-month study from Singapore with 32% showing high NAb level[12]. Our study further suggested, as shown in the 3-class model (though with worse statistical fit) in sensitivity analysis, similarly 15.5% of patients in Class 1 had persistence of high NAb for at least 9 months, while 47% of patients' NAb in Class 2A had declined to a moderately high level of about 70%, well above that of healthy adults in the moderate-waning class (Class 2B, 37.5%). The demographic and clinical characteristics of high-persistent and waning classes were similar, though high-persistent class' patients were older in age and more likely to have multiple chronic diseases instead of hypertension, diabetes and/or hyperlipidaemia per se. The higher and persistent level of NAb in elderly SARS-CoV-2 infection could be attributed to prolonged viral clearance and therefore continued immunological stimulation[20]. Likewise chronic medical conditions like hypertension, obesity and diabetes had been shown to be associated with prolonged virus clearance[21–23]. High virus burden might have led to strong and prolonged immunologic stimulation.

One potential implication of differentiation into classes is the provision of a framework for evaluating re-infection risks in the population. NAb levels have been shown to be predictive of immune protection from infection as well as severe disease[24]. Despite attainment of a moderate NAb level shortly after infection, waning class' patients (especially Class 2B in sensitivity analysis), comprising either satisfactory or mild conditions during hospitalization, stand high risk of re-infection towards the end of one year, especially that loss of NAb was beginning to emerge. Globally, SARS-CoV-2 re-infection has been increasingly reported[25], especially among highly exposed individuals like healthcare workers[26], and previously mild or asymptomatic patients[27]. While a sVNT threshold for immunological protection has yet to be established, vaccination at 6–9 months after onset would be beneficial. A recent study reported a higher odds of re-infection among the unvaccinated[28]. For patients in Class 2B, comprising younger adults with mild/moderate diseases, their NAb levels were continuing to decline at the end of 1 year. Vaccination of previously

infected patients not only gave a stronger neutralising response but could boost the neutralisation breadth conferred by natural infection[29]. Comprising largely elderly patients and people with multiple chronic illnesses, the high-persistent class is unique in the maintenance of NAb at relatively high level over time. At the end of one year, vaccination could further enhance protection from severe disease and the infection by SARS-CoV-2 variants in this vulnerable population[30]. Knowingly the responses of elderly people to vaccination may be less optimal[16], posing challenge in the development of optimal regimen for achieving effective long term prevention.

Our study carried some limitations. First, there was an uneven proportional distribution of recruited patients by class and by time. Specifically, the number of asymptomatic patients was smaller, and very few continued to attend clinic sessions beyond half year because of their generally healthy status. We were unable therefore to describe adequately the longitudinal NAb pattern of asymptomatic patients in this study. Nonetheless, under current isolate policy, reinfection would have been uncommon and the duration of hospitalisation was similar among four classes of patients. The sampling bias during hospitalisation has been reduced. Second, whereas sVNT levels are known to correlate with that of NAb[31], there's no established threshold for defining serological protection. In this study we have adopted a cut-off level of 20% inhibition for differentiating between positive and negative response. With only 2 patients with readings between 20% and 30%, our results remain robust if 30% is computed as reported in other studies[12,14]. Its relevance and implications in the continued expansion of variants in the virus population would need to be confirmed in longer term clinical studies. We are mindful also that the T-cell immunity may continue to be at work even with the absence of humoral responses to the virus[32]. Our study results are therefore meant for classification of patients into risk categories instead of specifying the level of immune responses for individuals. Third, our study has included convalescent patients recovering from COVID-19 while fatal cases were not represented. On the other end of the spectrum, asymptomatic patients tended to drop out from clinical care and were not adequately represented in the longitudinal assessment. Finally, our study was conducted when the prevailing virus carried D614G substitution[33], the epidemiology of which is different from that with subsequent emergence of variants including delta. The difference in the pattern of NAb responses between variants is not known, and the extrapolation of results should be cautioned.

In conclusion, we have classified COVID-19 patients into distinct classes by their longitudinal pattern of NAb responses

over a one-year period. The strength of the NAb responses and the persistence/waning patterns were largely associated with age, history of chronic illnesses and pre-morbid conditions. Whereas elderly people with chronic illnesses could achieve high persistent NAb level beyond 9 months, young and mildly symptomatic people had waning NAb level over time. The results could be useful for guiding the development of vaccination strategy for not just elderly individuals but convalescent COVID-19 patients belonging to waning class especially if high exposure risk in the population is anticipated.

## Data availability

The dataset cannot be deposited in a public repository because sharing of primary data is restricted by the Joint Chinese University of Hong Kong—New Territories East Cluster Clinical Research Ethics Committee due to confidentiality concerns on patient data. Anonymous clinical and laboratory data are owned by the Hospital Authority, while anonymous surveillance data are owned by the Department of Health, Hong Kong SAR Government, to which inquiry for access and permission for research purpose could be directed. However, the source data for the main figures are available as Supplementary Data 9.

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

## Acknowledgements

The authors thank Ms Vickie Li and Ms Carrie Yam for assistance in data management, and Mr. CY Poon and Ms. KY So for their assistance in the laboratory work. This research was supported by Health and Medical Research Fund Commissioned Research on the Novel Coronavirus Disease (COVID1903008-Project A, COVID190105), and Research Grants Council Collaborative Research Fund (C6107-20GF). The opinions and assertions contained herein are private views of the authors and do not necessarily reflect those of the Centre for Health Protection, Hong Kong Special Administrative Region Government Department of Health.

## Author contributions

N.S.W.: data analysis, original draft preparation, review and editing. S.S.L.: conceptualisation, original draft preparation, review and editing. D.P.C.C: laboratory processing and supervision, manuscript review and editing. T.C.M.L., T.H.Y.H., F.W.L.L., K.M.C., E.Y.K.T.: study preparation and execution, manuscript review and editing. E.K.Y., S.Y.S.W., D.S.C.H.: resources, manuscript review and editing. G.C.Y.L.: conceptualisation, study preparation and execution, original draft preparation, review and editing, resources.

## Competing interests

The authors declare no competing interests.
