## [Peer Review File · Communications Medicine]

Reviewers' comments:

Reviewer #1 (Remarks to the Author):

Thank you for your concise, important, and informative analysis. I have some minor questions and comments below which I hope will be helpful in your revisions.

Introduction: line 82 notes that the study aim was to examine trajectory patterns, but it would be helpful to list here exactly what would be examined (ie trajectory patterns of *what*).

Methods:

Study design – what was the time frame of the study? How does that relate to the introduction of vaccination in Hong Kong (ie assuring that the responses measured are due to infection and not vaccine) NOTE: I see now that this is in the results. I would normally have this in the methods so will leave my comment in but if no other reviewers mention this, then you can ignore my comment.

Participant section – it is not clear to me how asymptomatic persons were identified and recruited into the study. Were they hospitalized for other reasons with non-COVID-like symptoms but given a test? For someone outside of Hong Kong this might not be clear, if the approaches to who gets tested differ by country.

COVID serology – was the S ELISA introduced later? It is not clear why only the N ELISA and virus neutralization test were used to select people for the cohort.

Discussion:

Pg 7 it might be useful to separate out the discussion of Ct values and viral clearance, which start to get mixed up a bit. Theoretically there could be similar maximum Ct values in different individuals or groups but longer viral clearance for other reasons (immune response, treatment efficacy, etc).

Tracking VNT may be too challenging for some places – could the ELISA tests be used as a proxy for VNT? Was that analysis done at all? It isn't critical, but practically speaking, it would be great if we could use a simpler test to help identify people at risk or re-infection.

Reviewer #2 (Remarks to the Author):

Wong et al. examine the duration of SARS-CoV-2 neutralizing antibodies in 200 convalescent patients up to 14 months post symptom onset. The cohort consisted of individuals who were admitted to the hospital with a positive COVID nasopharyngeal PCR. Using these 200 patients who had 2 to 5 blood draws at different timepoints post infection, they classified the dynamics and duration of the SARS-CoV-2 neutralizing antibody response into 4 classes: high persistent, high waning, moderate waning, and low waning. The authors then characterized these classes of patients with individuals with high persistent nAb or high waning tended to be elderly with severe co-morbidities and with severe infection. These associations are not novel as had been observed in a cohort of 164 individuals followed for up to 6 months using a surrogate virus neutralization test (Chia et al. Dynamics of SARS-CoV-2 neutralising antibody responses and duration of immunity: a longitudinal study. *Lancet Microbe* 2021.) Another study noted waning nAb levels one year post symptom onset in 76 convalescent patients (Xiang et al. Declining Levels of Neutralizing Antibodies Against SARS-CoV-2 in Convalescent COVID-19 Patients One Year Post Symptom Onset. *Front.*

Immunol.) and had evaluated T cell responses. This work, however, appears to be the largest cohort to follow convalescent patients up to 14 months. This work expounds on the work of Chia et al. with a distinction between high persistent and high waning based on trajectory post 6 months.

The main point of the manuscript is the classification of patients based on neutralizing antibodies.

- Remove Table 1. The focus of the study should be on Table 2 and characterization of the patients based on their nAb classification.
- As there is plasma for these patients, complete the Spike IgG titers (EuroImmun) for this cohort.
- Figure 1 should show individual values versus months post symptom onset to help demonstrate the spread of patients over 14 months. Days post symptom onset better may be better to plot than months post symptom onset.
- Figure 3 should show the individual values for the 4 different classifications.
- What type of immunodeficiency did the 3 patients in the moderate-waning and low-waning nAb have? Did their immunodeficiency impact their ability to make a nAb response?
- Was there anything different about the 2 persons who seroreverted?
- The authors should show graphs for Spike IgG versus days post symptom onset and nucleocapsid IgG versus days post symptom onset with the individual values with the nAb classifications of these individuals as additional figures for Figure 3.

Reviewer #3 (Remarks to the Author):

Introduction: 'In the absence of effective curative therapy, recovery of SARS-CoV-2 infected patients depends almost exclusively on the functioning of host immunity'. This statement isn't correct, vaccines protect most infected individuals really well against severe disease. Furthermore, effective therapies have been recently been approved in at least some countries, such as Lagevrio in the UK. Several other drugs are being used in practice, e.g. see a simply summary:

<https://www.health.harvard.edu/diseases-and-conditions/treatments-for-covid-19>

A recent study from the UK, including milder cases that were not hospitalised, that used very similar statistical methodologies could also be discussed in the introduction or discussion:

<https://www.nature.com/articles/s41467-021-26479-2>. It would also be appropriate to describe that LCMM has been used before in other studies to classify (anti-spike) antibody trajectories, e.g.

<https://www.nature.com/articles/s41467-021-26479-2>; <https://www.nature.com/articles/s41564-021-00947-3>.

Why are only default equidistant splines evaluated? This may well lead to suboptimal number and placement of knots. It is straightforward to use custom splines, e.g. see:

<https://www.nature.com/articles/s41467-021-26479-2>

Why adjust for age in such a crude binary way (60+ or below)? This assumes that there is a sudden shift at age of 60, which is biologically implausible. A priori, an e.g. restricted cubic spline is expected to lead to a better model fit that is more consistent with underlying biology.

How many classes were considered for the latent class model? It's unclear whether models with different number of classes were tested. Most classes are really small (29-39 patients), and I wonder

whether the authors are not overfitting. Furthermore, I can imagine that with low numbers, the actual classes and who ends up in which class is quite sensitive to what kind of spline is being used.

When referring to ORs in the main text, please also report the confidence intervals.

Complete-case analyses were performed, but how many missing values were there? I.e. is this potentially problematic or is most data complete anyway?

What are the sensitivity and specificity of the used tests? Are some of the trajectories simply a consequence of random false positives / negatives?

The discussion claims that one important implication of differentiation into classes is the provision of an evidence-based framework for assessing re-infection risks in the population. However, it is not clear (yet) from the results and text that this can be genuinely claimed. NAb levels have been predictive of immune protection from infection, but it's not immediately apparent whether this is relates to peak or also decline. If the peak level is by far most important, classifying trajectories wouldn't be really important and if you have a measurement from someone at a given point in time you may not be able to say much if you don't know the peak level in that case.

Appendix table 1: The age for both groups isn't provided, despite it being listed as a variable in this table. Furthermore, exact p-values should be provided (up to a certain level of precision e.g. 0.001), not just a * indicating a p-value <0.05. It also reads better if p-values are provided in a separate column and not together with the OR and 95% CI. Furthermore, it is unclear why it is of interest to test for difference in baseline characteristics for these two age-groups.

Appendix table 3. Please specify that this is obtained using a latent class logistic regression model and not a standard logistic regression. Please also provide posterior class-membership probabilities in the table.

COMMSMED-21-0484-T

Reviewers' comments:

Reviewer #1 (Remarks to the Author):

Thank you for your concise, important, and informative analysis. I have some minor questions and comments below which I hope will be helpful in your revisions.

COMMENT: Introduction: line 82 notes that the study aim was to examine trajectory patterns, but it would be helpful to list here exactly what would be examined (ie trajectory patterns of *what*).

RESPONSE: It has been revised as “It is against this background that we undertook to examine the trajectory of neutralising antibody (NAb) responses and factors associated with its pattern in SARS-CoV-2 patients who had recovered from COVID-19 of different severity.”

Methods:

COMMENT: Study design – what was the time frame of the study? How does that relate to the introduction of vaccination in Hong Kong (ie assuring that the responses measured are due to infection and not vaccine) NOTE: I see now that this is in the results. I would normally have this in the methods so will leave my comment in but if no other reviewers mention this, then you can ignore my comment.

RESPONSE: Thanks for your suggestion. We have revised the METHODS, under study design and data sources: “This is a longitudinal study with the collection of blood samples and clinical data in COVID-19 patients between February 2020 and February 2021 in Hong Kong, where stringent isolation policy is in place with all confirmed cases are reported and hospitalised irrespective of symptoms, travel, and contact history. COVID-19 vaccination programme started from 22 February 2021 while the reporting date of the last recruited patient was 20 February 2021.”

COMMENT: Participant section – it is not clear to me how asymptomatic persons were identified and recruited into the study. Were they hospitalized for other reasons with non-COVID-like symptoms but given a test? For someone outside of Hong Kong this might not be clear, if the approaches to who gets tested differ by country.

RESPONSE: In Hong Kong, stringent isolation policy is in place requiring all confirmed cases to be reported and hospitalised irrespective of symptoms, travel, and contact history. Through screening programme and contact tracing, asymptomatic cases were identified when they were tested positive with RT-PCR. SARS-CoV-2 infected patients were discharged on recovery and when they tested twice negative for PCR and/or positive for antibody in accordance with the prevailing hospital guideline.

COMMENT: COVID serology – was the S ELISA introduced later? It is not clear why only the N ELISA and virus neutralization test were used to select people for the cohort.

RESPONSE: The sentence (p.5, general characteristics) has been amended to include IgG anti-spike, though not all samples have been tested for the 3 tests at all time-points, as limited by specimen availability: “Of these, 368 with IgG NP, IgG spike and NAb measurements in day 15-90 were selected for analysis.”

Discussion:

COMMENT: Pg 7 it might be useful to separate out the discussion of Ct values and viral clearance, which start to get mixed up a bit. Theoretically there could be similar maximum Ct values in different individuals or groups but longer viral clearance for other reasons (immune response, treatment efficacy, etc).

RESPONSE: The discussion on Ct values has been removed so as not to mix up with that on viral clearance. The description on Ct value is added back to the last paragraph of the RESULTS.

COMMENT: Tracking VNT may be too challenging for some places – could the ELISA tests be used as a proxy for VNT? Was that analysis done at all? It isn’t critical, but practically speaking, it would be great if we could use a simpler test to help identify people at risk or re-infection.

RESPONSE: We agree that IgG anti-spike would likely be a proxy for sVNT. Recent studies suggested that correlation between the 2 forms of tests was good but comparability between different anti-spike antibody assays has remained a problem. The application of IgG anti-spike in place of sVNT (itself a surrogate for NAb test) would only be possible when a standard test and cut-off is established. As this subject falls outside the scope of the current study, this has not been included in the DISCUSSION.

Reviewer #2 (Remarks to the Author):

Wong et al. examine the duration of SARS-CoV-2 neutralizing antibodies in 200 convalescent patients up to 14 months post symptom onset. The cohort consisted of individuals who were admitted to the hospital with a positive COVID nasopharyngeal PCR. Using these 200 patients who had 2 to 5 blood draws at different timepoints post infection, they classified the dynamics and duration of the SARS-CoV-2 neutralizing antibody response into 4 classes: high persistent, high waning, moderate waning, and low waning. The authors then characterized these classes of patients with individuals with high persistent nAb or high waning tended to be elderly with severe co-morbidities and with severe infection. These associations are not novel as had been observed in a cohort of 164 individuals followed for up to 6 months using a surrogate virus neutralization test (Chia et al. Dynamics of SARS-CoV-2 neutralising antibody responses and duration of immunity: a longitudinal study. Lancet Microbe 2021.)

Another study noted waning nAb levels one year post symptom onset in 76 convalescent patients (Xiang et al. Declining Levels of Neutralizing Antibodies Against SARS-CoV-2 in Convalescent COVID-19 Patients One Year Post Symptom Onset. Front. Immunol.) and had evaluated T cell responses. This work, however, appears to be the largest cohort to follow convalescent patients up to 14 months. This work expounds on the work of Chia et al. with a distinction between high persistent and high waning based on trajectory post 6 months.

The main point of the manuscript is the classification of patients based on neutralizing antibodies.

COMMENT: - Remove Table 1. The focus of the study should be on Table 2 and characterization of the patients based on their nAb classification.

RESPONSE: Table 1 has been relocated to Appendix Table for the reference of readers who would like to have a fuller picture about the general characteristics of enrolled patients.

COMMENT: - As there is plasma for these patients, complete the Spike IgG titers (EuroImmun) for this cohort.

RESPONSE: As suggested, IgG Spike results (expressed in ratio) have been summarized with median and IQR over time periods in Appendix Table 3.

COMMENT: - Figure 1 should show individual values versus months post symptom onset to help demonstrate the spread of patients over 14 months. Days post symptom onset better may be better to plot than months post symptom onset.

RESPONSE: Figure 1 has been revised accordingly focusing on the symptomatic patients.

COMMENT: - Figure 3 should show the individual values for the 4 different classifications.

RESPONSE: We have added individual values for 4 different classifications on Figure 3.

COMMENT: - What type of immunodeficiency did the 3 patients in the moderate-waning and low-waning nAb have? Did their immunodeficiency impact their ability to make a nAb response?

RESPONSE: They were all HIV patients. The figure below compares between patients with and without immunodeficiency. The response was similar between the two groups. However, with the limited number of immunodeficiency patients and small number of followup time points, a definitive conclusion cannot be drawn.

COMMENT: - Was there anything different about the 2 persons who seroreverted?

RESPONSE: The two patients were different in terms of age.

Case #104: Male aged 64, symptomatic, NAb (%) from symptom onset date was 38.8% (30 days) and 16% (326 days). He had allergic rhinitis with long-standing post-nasal drip, chronic cough, long term history of sputum. His clinical disease was of moderate severity, and was hospitalised for 10 days.

Case #363: Male aged 23, symptomatic, NAb (%) from symptom onset date was 58.4% (47 days), 50.1% (103 days), 32.3% (182 days), and 6% (378 days). He had good past health. His clinical disease was mild, and was hospitalised for 6 days.

COMMENT: - The authors should show graphs for Spike IgG versus days post symptom onset and nucleocapsid IgG versus days post symptom onset with the individual values with the nAb classifications of these individuals as additional figures for Figure 3.

RESPONSE: A figure (Appendix Figure 2) showing IgG spike ratio versus days from symptom onset with NAb classifications are presented. IgG NP values are not shown as the results were expressed as either positive or negative.

Reviewer #3 (Remarks to the Author):

COMMENT: Introduction: 'In the absence of effective curative therapy, recovery of SARS-CoV-2 infected patients depends almost exclusively on the functioning of host immunity'. This statement isn't correct, vaccines protect most infected individuals really well against severe disease. Furthermore, effective therapies have been recently been approved in at least some countries, such as Lagevrio in the UK. Several other drugs are being used in practice, e.g. see a simply summary: <https://www.health.harvard.edu/diseases-and-conditions/treatments-for-covid-19>

RESPONSE: The sentence has been updated as “Before the widespread use of vaccine and in the absence of definitive curative therapy in most places, recovery of SARS-CoV-2 infected patients depends almost exclusively on the functioning of host immunity.”

COMMENT: A recent study from the UK, including milder cases that were not hospitalised, that used very similar statistical methodologies could also be discussed in the introduction or discussion: <https://www.nature.com/articles/s41467-021-26479-2>. It would also be appropriate to describe that LCMM has been used before in other studies to classify (anti-spike) antibody trajectories, e.g. <https://www.nature.com/articles/s41467-021-26479-2>; <https://www.nature.com/articles/s41564-021-00947-3>.

RESPONSE: Thanks for your suggestion. We have added the reference in METHODS “LCMM is a combination of latent class model and mixed model for repeated measurements, which have been applied in classifying the anti-spike antibody response to SARS-CoV-2 vaccines,[16] and natural SARS-CoV-2 infection [17].”

COMMENT: Why are only default equidistant splines evaluated? This may well lead to suboptimal number and placement of knots. It is straightforward to use custom splines, e.g. see: <https://www.nature.com/articles/s41467-021-26479-2>

RESPONSE: With reference to the suggested article, we have tried the custom splines with knots in months 1, 2 and 3 in 1 model, and months 1, 3 and 6 in the another model, whose BIC was however higher than the model selected in the manuscript (the summary table below refers). We have therefore opted to adopt the original model.

COMMENT: Why adjust for age in such a crude binary way (60+ or below)? This assumes that there is a sudden shift at age of 60, which is biologically implausible. A priori, an e.g. restricted cubic spline is expected to lead to a better model fit that is more consistent with underlying biology.

RESPONSE: We did not adjust for age in LCMM. We have instead examined the variations between identified LCMM classes and described the different characteristics in each class including age.

COMMENT: How many classes were considered for the latent class model? It's unclear whether models with different number of classes were tested. Most classes are really small (29-39 patients), and I wonder whether the authors are not overfitting. Furthermore, I can imagine that with low numbers, the actual classes and who ends up in which class is quite sensitive to what kind of spline is being used.

RESPONSE: We have considered 2, 3, and 4 classes, under link functions of 3-equi-splines, 5-equi-splines, and the suggested custom knots. BIC was the lowest in 5-equi-splines models, with 4 classes having better posterior probabilities above a threshold of 70% for all classes. Below please find the summary of model results and we have included this in Appendix Table 4:

	2 classes model	3 classes model	4 classes model
3-equi-splines			
BIC	3871.78	3876.05	3881.64
Posterior probabilities above a threshold of 70%	99.47; 72.73	82.83; 86.96; 93.75	50.00; 77.78; 87.95; 93.94
5-equi-splines			
BIC	3838.55	3843.73	3847.03
Posterior probabilities above a threshold of 70%	70.97; 95.27	74.47; 83.87; 74.67	74.36; 86.21; 87.63; 91.43
Custom knots at months 1, 2 and 3			
BIC	4083.68	4048.22	4072.74
Posterior probabilities above a threshold of 70%	87.69; 94.81	77.42; 80 91.47	82.76; 69.05; 35.56; 48.81
Custom knots at months 1, 3 and 6			
BIC	4050.01	4051.29	4072.93
Posterior probabilities above a threshold of 70%	93.94; 100	92.86; 81.48; 94.07	89.66; 72.92; 39.13; 54.55

COMMENT: When referring to ORs in the main text, please also report the confidence intervals.

RESPONSE: Confidence intervals have been added in the main text.

COMMENT: Complete-case analyses were performed, but how many missing values were there? I.e. is this potentially problematic or is most data complete anyway?

RESPONSE: For 200 patients included in trajectory classification analysis, only 8 had missing values for ethnicity (Chinese vs non-Chinese), and 6 missed values for all laboratory measurements on admission and during hospitalisation but 85 had missing values for Globulin alone, and 92 had missing values for IgG Spike within 3 months. The number of missing values has been added to the Tables.

COMMENT: What are the sensitivity and specificity of the used tests? Are some of the trajectories simply a consequence of random false positives / negatives?

RESPONSE: Using sVNT at 20% inhibition cutoff, the sensitivity was 91.2% (81.8-96.7%) and the specificity was 94.4% (89.2-97.5%) among samples collected >14 days post symptom onset (Bond K, et al. Evaluation of Serological Tests for SARS-CoV-2: Implications for Serology Testing in a Low-Prevalence Setting. J Infect Dis. 2020;222:1280-1288.).

In the trajectory analysis, we have used sVNT inhabitation (%) instead of positive and negative value (binary), which has not affected the trajectory classification.

COMMENT: The discussion claims that one important implication of differentiation into classes is the provision of an evidence-based framework for assessing re-infection risks in the population. However, it is not clear (yet) from the results and text that this can be genuinely claimed. NAb levels have been predictive of immune protection from infection, but it's not immediately apparent whether this relates to peak or also decline. If the peak level is by far most important, classifying trajectories wouldn't be really important and if you have a measurement from someone at a given point in time you may not be able to say much if you don't know the peak level in that case.

RESPONSE: Mindful of the uncertainties, we have revised the sentence to tone down the claim. "One potential implication of differentiation into classes is the provision of a framework for evaluating re-infection risks in the population."

COMMENT: Appendix table 1: The age for both groups isn't provided, despite it being listed as a variable in this table. Furthermore, exact p-values should be provided (up to a certain level of precision e.g. 0.001), not just a * indicating a p-value <0.05. It also reads better if p-values are provided in a separate column and not together with the OR and 95% CI. Furthermore, it is unclear why it is of interest to test for difference in baseline characteristics for these two age-groups.

RESPONSE: Appendix Table 1 has been revised accordingly. We compared the characteristics between the two age groups as age was likely to be a significant factor associated with the NAb trajectory pattern.

COMMENT: Appendix table 3. Please specify that this is obtained using a latent class logistic regression model and not a standard logistic regression. Please also provide posterior class-membership probabilities in the table.

RESPONSE: Results in appendix Table 3 were based on simple logistic regression model instead of latent class logistic regression model. We used the class identity from LCMM as outcome variables. Our apology for the confusion. We have added a self-explanatory sentence beneath the table title.

Posterior class-membership probabilities have been added in Table 1 Comparison of characteristics by class in latent class mixed model (LCMM) in the main text.

Reviewers' comments:

Reviewer #2 (Remarks to the Author):

The authors have addressed the concerns in the initial view of their manuscript. The authors suggest that their classification of nAb responses can predict risk for re-infection. Individuals who were asymptomatic are less likely to retain high nAb titer and therefore are at risk for reinfection. Their last sentence is somewhat counterintuitive -- younger individuals (asymptomatic or mildly symptomatic) would need to be prioritized over the elderly (more symptomatic/severe symptoms). The statement reflects their data but is counterintuitive to that of public health departments -- prioritizing the elderly over the young.

For Figure 3, please format the x-axis as days post symptom onset instead of months post symptom onset to match Figure 1.

For future directions, what does hybrid immunity (infection + vaccination) look like in your cohort and how does that impact your classification system?

Reviewer #3 (Remarks to the Author):

The authors have addressed most comments. I do, however, have one remaining concern:

- the best fitting model according to the BIC is the 2-class model, as such it is unclear why the authors choose to present the 4-class model as the main result, especially given the low number of patients in each class. This seems quite arbitrary and I think the 2-class model should be the main analyses given that the BIC suggests this model is better than the 4-class model. Obviously this would substantially change the findings as there is only sufficient data to support 2 distinct classes.

COMMSMED-21-0484A

Reviewers' comments:

Reviewer #2 (Remarks to the Author):

COMMENT: The authors have addressed the concerns in the initial view of their manuscript. The authors suggest that their classification of nAb responses can predict risk for re-infection. Individuals who were asymptomatic are less likely to retain high nAb titer and therefore are at risk for reinfection. Their last sentence is somewhat counterintuitive -- younger individuals (asymptomatic or mildly symptomatic) would need to be prioritized over the elderly (more symptomatic/severe symptoms). The statement reflects their data but is counterintuitive to that of public health departments -- prioritizing the elderly over the young.

RESPONSE: We have rephrased the last sentence to highlight the need for development vaccination strategy for convalescent COVID-19 patients, the prioritisation of which would need to be considered separately from addressing the infection risk of the elderly:

“The results could be useful for guiding the development of vaccination strategy for not just elderly individuals but convalescent COVID-19 patients belonging to moderate-waning and low-waning classes especially if high exposure risk in the population is anticipated.”

COMMENT: For Figure 3, please format the x-axis as days post symptom onset instead of months post symptom onset to match Figure 1.

RESPONSE: The x-axis of Figure 3 has been revised as suggested.

COMMENT: For future directions, what does hybrid immunity (infection + vaccination) look like in your cohort and how does that impact your classification system?

RESPONSE: With the emergence of variants and the introduction of booster (third) vaccine doses, future interpretation would need to take into consideration the factors of vaccination coverage of (full course +/- booster/3rd dose) and the distribution of the variants. The classes would likely remain similar as they are derived from the trajectory patterns while the implications for interventions may vary. The Research Team is in the process of developing follow-up study.

Reviewer #3 (Remarks to the Author):

The authors have addressed most comments.

I do, however, have one remaining concern:

COMMENTS: - the best fitting model according to the BIC is the 2-class model, as such it is unclear why the authors choose to present the 4-class model as the main result, especially given the low number of patients in each class. This seems quite arbitrary and I think the 2-class model should be the main analyses given that the BIC suggests this model is better than the 4-class model.

Obviously this would substantially change the findings as there is only sufficient data to support 2 distinct classes.

RESPONSE: In the 2-class model, the distribution of class 1 was 15.5% (31 cases) and class 2 was 84.5% (169 cases). The interpretability of 2-class model is doubtlessly lower for such a model despite the slightly lower BIC than the 4-class model (5-equi-splines: 3847.03 for 4-class, 3838.55 for 2-class). The decision of a 4-class model has been discussed by the Research Team in terms of statistical power and clinical interpretability. We have added the posterior classification in Appendix Table 4 for the reference for readers. The criteria for model selection has been updated in the main text, Methods (p.5): "Model with lower Bayesian information criterion (BIC) value and better clinical interpretability was selected"

Reviewers' comments:

Reviewer #3 (Remarks to the Author):

I am sorry, but I strongly disagree with favouring the 4 model based on 'clinical interpretability'. A common problem with agnostic latent class models is that it is often easy to explain - on hindsight - a higher number of classes than the data support.

If anything, it is better practice to be on the cautious side with latent class models and select a model with fewer classes, which is in virtually all applications easier to interpret and work with.

The data does simply NOT support the 4 class model - which results in relatively small groups, highlighting the potential issue with overfitting as also indicated by the higher BIC of this model - over the 2 class model.

I am not saying there are truly no 4 classes, but the current data does not provide evidence that this is the case. As such it is statistically not justified to use 4 classes and the paper should really be rewritten using 2 classes or additional data needs to be included/collected which may or may not lead to a 4 class model being selected.

COMMSMED-21-0484B

Reviewer #3 (Remarks to the Author):

COMMENT: I am sorry, but I strongly disagree with favouring the 4 model based on 'clinical interpretability'. A common problem with agnostic latent class models is that it is often easy to explain - on hindsight - a higher number of classes than the data support.

If anything, it is better practice to be on the cautious side with latent class models and select a model with fewer classes, which is in virtually all applications easier to interpret and work with.

The data does simply NOT support the 4 class model - which results in relatively small groups, highlighting the potential issue with overfitting as also indicated by the higher BIC of this model - over the 2 class model.

I am not saying there are truly no 4 classes, but the current data does not provide evidence that this is the case. As such it is statistically not justified to use 4 classes and the paper should really be rewritten using 2 classes or additional data needs to be included/collected which may or may not lead to a 4 class model being selected.

RESPONSE: We understand reviewer's concern and have revised the analysis and results using 2-class model. Discussion part has been revised in accordance with the modified results. While we agree that statistically 2-class model was a better option, please accept our inclusion of other class models' results as sensitivity analysis (details in the Appendix) for exploring the potential pattern of sVNT trajectories.

Reviewers' comments:

Reviewer #3 (Remarks to the Author):

I'm happy with the changes, and the manuscript can be accepted for publication conditional on the following change/clarification (taking into account that there is no strong statistical support for the model in the sensitivity analysis):

'Our study further demonstrates the persistence of high NAb in 15.5% of these patients for at least 9 months, while the level of the other 47% (Class 2A) declined to a moderately high level of about 70%, well above that of healthy adults in the moderate-waning class (Class 2B) as shown in the 3-class model in sensitivity analysis.'

Please avoid the word demonstrates, and use suggests instead.

Also the sentence is not completely clear to me. Do high NAb persist in 15.5% of 15% patients (further above in same alinea)? Similarly, what is the other 47% (suggests that 53% of patients are somewhere else, but you only have mentioned 15% so far (and it reads as if you say 15% of 15.5%).

Please clarify this better and highlight in this sentence that the 3-class model had worse statistical fit, it is important to caveat that here in the discussion again.

COMMSMED-21-0484C

Reviewer #3 (Remarks to the Author):

I'm happy with the changes, and the manuscript can be accepted for publication conditional on the following change/clarification (taking into account that there is no strong statistical support for the model in the sensitivity analysis):

'Our study further demonstrates the persistence of high NAb in 15.5% of these patients for at least 9 months, while the level of the other 47% (Class 2A) declined to a moderately high level of about 70%, well above that of healthy adults in the moderate-waning class (Class 2B) as shown in the 3-class model in sensitivity analysis.'

Please avoid the word demonstrates, and use suggests instead.

RESPONSE: The word “demonstrates” has now been replaced by “suggests”.

Also the sentence is not completely clear to me. Do high NAb persist in 15.5% of 15% patients (further above in same alinea)? Similarly, what is the other 47% (suggests that 53% of patients are shomewhere else, but you only have mentioned 15% so far (and it reads as if you say 15% of 15.5%).

Please clarify this better and highlight in this sentence that the 3-class model had worse statistical fit, it is important to caveat that here in the discussion again.

RESPONSE: The 15.5% in the 2-class model reflecting high persistent response was similar to the 15.5% in the 3-class model. The description did not mean 15.5% of 15% patients, while the other 47% referred to Class 2A in the 3-class model, with the reminder of 37.5% belonging to Class 2B. The two sentences have been amended for clarification, and the worse statistical fit for the 3-class model specified:

“Accounting for 15.5% of the study population, the high-persistent class in the 2-class model comprised mainly elderly patients with chronic illnesses who, following SARS-CoV-2 infection, could generate high levels of NAb. Our results echoed that of a shorter 6-month study from Singapore with 32% showing high NAb level.[12] Our study further suggested, as shown in the 3-class model (though with worse statistical fit) in sensitivity analysis, similarly 15.5% of patients in Class 1 had persistence of high NAb for at least 9 months, while 47% of patients' NAb in Class 2A had declined to a moderately high level of about 70%, well above that of healthy adults in the moderate-waning class (Class 2B, 37.5%).”